# Automatically Constructing Compositional and Recursive Learners

## Abstract

We consider learning to generalize and extrapolate with limited data to harder compositional problems than a learner has previously seen. We take steps toward this challenge by presenting a characterization, algorithm, and implementation of a learner that programs itself automatically to reflect the structure of the problem it faces. Our key ideas are (1) transforming representations with modular units of computation is a solution for decomposing problems in a way that reflects their subproblem structure; (2) learning the structure of a computation can be formulated as a sequential decision-making problem. Experiments on solving various multilingual arithmetic problems demonstrate that our method generalizes out of distribution to unseen problem classes and extrapolates to harder versions of the same problem. Our paper provides the first element of a framework for learning general-purpose, compositional and recursive programs that design themselves.

## 1. Introduction

Teach a human to fish and feed them for a lifetime. Teach a machine to stack 3 blocks and it still needs to be taught how to stack 4 blocks. And 5 blocks. And 6 blocks. While machine learning has seen recent success in mastery of specific individual skills, it is still a challenge for machines to reuse past experience to solve harder problems than it has seen, especially in low data regimes. This is because many methods do not focus on learning algorithmic procedures that take advantage of the compositional structure in problems. How can we enable a learner to capture the essence of what makes a hard problem more complex than a simple one, break the hard problem along characteristic lines into smaller problems it knows how to solve, and sequentially solve the smaller problems until the larger one is solved?

[1]Anonymous Institution, Anonymous City, Anonymous Region, Anonymous Country. Correspondence to: Anonymous Author <anon.email@domain.com>.

Preliminary work. Under review by the International Conference on Machine Learning (ICML). Do not distribute.

To make progress towards this goal, this paper focuses on learning programs for solving a particular family of problems that exhibit compositional structure: their solutions can be found by composing in sequence a small set of reusable partial solutions. Such a family offers several attractive challenges. Firstly, we face the *translational challenge* which requires the learner to solve problems of a different category than it has seen, in which case it must learn to make an analogy between the new problem and previously encountered problems by re-representing the new problem into a category it is familiar with. We also face the *recursive challenge* which requires the learner to solve problems of the same category but are strictly harder, in which case it must learn an algorithm to iteratively reduce the harder problem into simpler problems of the same category.

The key idea behind this paper is to reformulate both analogy making and algorithm design as a sequential decision-making problem over transformations between representations. Unlike neural program synthesis methods that prespecify the transformations and learn the structure with supervised learning (Reed & De Freitas, 2015; Cai et al., 2017) and reinforcement learning (Chen et al., 2017; Ganin et al., 2018; Bunel et al., 2018), and unlike neural program induction methods that learn the transformations but prespecify the structure (Devin et al., 2017; Andreas et al., 2016; Riedel et al., 2016), our learner learns, with sparse supervision, both the structure and transformations of a modular recursive program that iteratively re-represents the input representation into more familiar representations that it knows how to compute with. One advantage is that the complexity of our program can dynamically be customized to the particular problem instance. Another advantage is that, by drawing the boundaries of modularity via transformations between representations, the program can learn to re-represent a new problem into one it already knows how to solve, rather than learning to solve it from scratch.

To operationalize our method in neural networks, we observe that a feedforward pass through a neural network is essentially the execution of an algorithm. This algorithm is a sequence of computations, where a computation transforms one representation to another representation. To learn such an algorithm, we further observe that the execution of an algorithm can be seen as a traversal through a Markov decision process (MDP), where the actions are the computations

and the states are intermediate results. Making this connection between programs and MDPs opens opportunities to borrow tools from reinforcement learning as a practical means for solving the discrete optimization problem of selecting which transformation to apply next.

The closest work to ours are Rosenbaum et al. (2017) and Gaunt et al. (2016). We build upon Rosenbaum et al. (2017) by removing the restriction of pre-specified execution length and allow function reuse within the same execution trace. Whereas Gaunt et al. (2016) assumes a differentiable interpreter grounded in source code, we take a different perspective of learning algorithms through re-representations, which enables our method to adapt on-line to problem categories it has not seen before.

## 2. General Setup

To solve a problem means transforming its input representation $x_i$ into its output representation $y_i$. $x_i$ and $y_i$ are associated with their respective *types* $t_x$ and $t_y$. In a *translational problem*, the input representation and the output representation differ (i.e. $t_x \neq t_y$) and in a *recursive problem*, they are the same (i.e. $t_x = t_y$). The *translational challenge* requires solving harder translational problems, which have unseen pairs of input-output types (e.g., the learner was trained to translate French to English and English to Spanish but not French to Spanish). The *recursive challenge* requires solving harder recursive problems, which require more computation (e.g., the learner was trained to solve up to 5-length arithmetic expressions but is tested on 10-length expressions). To extrapolate to harder problems, the learner must learn to discover and exploit the compositional structure of problems. However, in general the internal compositional structure of problems is unknown and the learner only has access to the input and desired output. For a particular problem $P_i$, the learner is given an input and the output type $(x_i, t_{y_i})$ and is expected to produce output $y_i$. The only supervision signal it receives is its prediction error. How does a learner automatically program itself to reflect the structure of each problem?

## 3. A Learner That Programs Itself

We present three main ideas. The first is that transforming representations with modular units of computation is a way to decompose problems to reflect their subproblem structure, yielding desirable generalization and extrapolation capabilities. The second formulates learning the structure of a compositional problem as a sequential decision-making process, yielding an algorithm for achieving the solution. The third is a method for implementing this algorithm in learnable, differentiable programs that make it possible to learn the underlying computational units.

### 3.1. Pattern-matching over problems via analogy

The first observation we make is that if the learner is trained on a diverse distribution of compositional problems that share enough structure it can distill out the individual structural components into specialized, modular *computational units*: atomic function operators that perform transformations between representations. Indeed, it is often the case that it is easier to solve a problem $P_a$ by making an analogy to a problem $P_b$ that one already knows how to solve, rather than retraining to solve $P_b$ from scratch (Schmidhuber, 2015). Therefore, if a learner hasn't ever translated French to Spanish but knows how to translate French to English and English to Spanish, it can reduce the French to Spanish problem to a English to Spanish problem by transforming the French input to English input, which it knows how to compute with. If a learner hasn't ever seen a 10-length arithmetic expression before, but it knows how to reduce an $n$-length expression to an $(n-1)$-length one, then it can iteratively reduce the 10-length expression down to a length it knows. Specialization naturally emerges when the computational units do not know the global problem they are solving but can make local progress that iteratively re-represents the problem into a more familiar form.

### 3.2. Learning the structure of a computation as a sequential decision-making process

Our second observation is that a transformation between representations can be generalized as any computation which changes the state of a program to another. Therefore, learning how to apply computational units in sequence can then be formulated as a sequential decision-making problem, where the state space is the intermediate results produced by a program and the action space is the set of computations.

This kind of sequential decision problem can be formalized as a meta-level Markov decision process (MDP) (Hay et al., 2014), defined by a tuple $(\mathcal{X}, \mathcal{F}, \mathcal{P}_{meta}, r, \gamma)$. $\mathcal{X}$ is the set of information states (intermediate results of computation), $\mathcal{C}$ is the set of computations, $\mathcal{P}_{meta}(x_j, f_j, x_{j+1})$ is the transition model that expresses the probability that at step $j$ the computation $f_j$ will change the information state from $x_j$ to $x_{j+1}$, $\gamma$ is a discount factor. The goal of the learner is to select a series of computations $f$ to iteratively transform the input $x$ into its predicted output $\hat{y}$. When the learner selects a special computation, the `halt` signal, the current result is produced as output. The learner incurs a cost for every computation it executes and receives a terminal reward that reflects how its output $\hat{y}$ matches the desired output $y$. The computation cost and the `halt` signal differentiate the meta-level MDP from a generic MDP, requiring the learner to balance program complexity and performance. The result is a program composed of a sequence of computations that customizes its complexity to the problem.

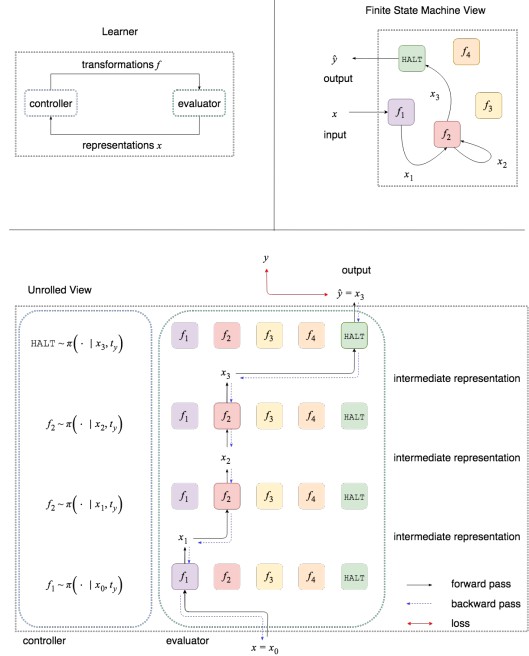

Figure 1. **Overview:** *top-left*: The learner is a cycle between a controller and evaluator: the controller selects a function $f$ given an intermediate representation $x$ and the evaluator applies $f$ on $x$ to create a new representation. *top-right*: The learner dynamically learns the structure of a program (viewed as a finite state machine) customized for its problem. *bottom*: A series of computations in the program is equivalent to a traversal through the Meta-MDP, where functions can be reused across different stages of computation, allowing for recursive computation.

### 3.3. A general practical algorithm for learning and composing computations

The meta-level MDP describes how to choose computations, but not how to learn the form of the computations themselves. We solve this problem by learning differentiable programs composed of neural networks, in which both the computations and their composition are learned.

The implementation (Figure 1) consists of a controller $\pi(f|x, t_y)$, a set of functions $f_k \in \mathcal{F}$, and an evaluator. At step $j$, the controller observes the intermediate state of computation $x_j$ and the target output type $t_y$ and selects a function $f_k$. The evaluator then applies $f_k$ to transform $x_j$ into a new intermediate state of computation $x_{j+1}$. When $\pi$ selects the `halt` signal, a loss is computed by comparing the current state of computation with the desired output. The loss is backpropagated through the functions, which are trained with Adam (Kingma & Ba, 2014). The controller receives a sparse reward derived from the loss, incurs a cost for every computation, and is trained with Proximal Policy Optimization (Schulman et al., 2017).

The state space (excluding the initial state, which is the input

to the learner) consists of the outputs of the functions, the action space consists of the functions themselves, and the transition model is the evaluator. As the learner trains, the function parameters change, so transition dynamics are non-stationary. Because the outputs of a function are the internal representations of the larger neural network, the learner simultaneously designs its own internal representation language and the transformations that convert between them. These internal representations may or may not correspond with the external representation given by the problem.

## 4. Experiments

Our experiments are aimed at evaluating the generalization, extrapolation, and compositional learning capabilities of our approach. We test on the translational and the recursive challenges. Specifically: (1) Can we generalize to unseen problem categories using combinations of transformations learned from prior tasks? (2) Can we generalize to harder problem instances, i.e. extrapolate?

**Multilingual Arithmetic Task:** Although our method can be applied to various multitask learning settings, multilingual arithmetic is a rich family of tasks that exhibit the properties we desire: (a) variable length inputs, (b) variable problem complexity, (c) reappearance of subproblems across problem, (d) reappearance of subproblems within a single problem, (e) a large combinatorial space created by composing together a small set of primitives using simple rules, (f) a natural curriculum of easier (shorter) to harder (longer) problems (g) diversity in the semantic roles of primitive components (e.g. operators like $+$, $\times$) (h) mulitmodality in representations. Any other high-dimensional, continuous input task would at least have the structure we study here in arithmetic; understanding how to learn to decompose and exploit the structure of arithmetic is a necessary prerequisite to solve any more complex task domain.

Multilingual arithmetic problems (modulo 10) come in $m$ different *types*, each corresponding to a language that expresses it. An example 3-term input expression is `three plus four times seven` (the *source* language is English) and an example desired output is `uno` (the *target* language is Spanish). The learner must be able to translate between types as well as solve the arithmetic problem. We arbitrarily chose $m = 5$ languages: English, Numerals, PigLatin, Reversed-English, Spanish. During training, each source language is seen with four target languages (and one held out for testing) and each target language is seen with four source languages (and one held out for testing). Our functions consist of "reducers" which reduce a length $k$ expression to a length $k - 3$ one: the controller chooses a window of 3 terms in the expression, and the reducer transforms that into a softmax distribution over a single term in the vocabulary; and "translators", which produces a length $k$

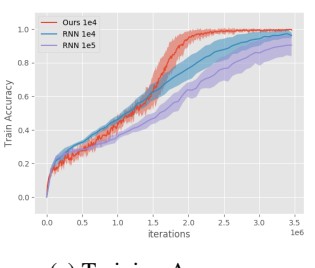

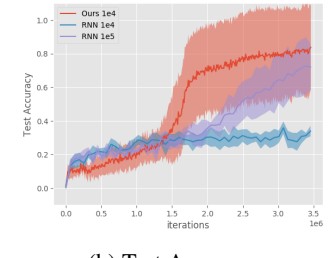

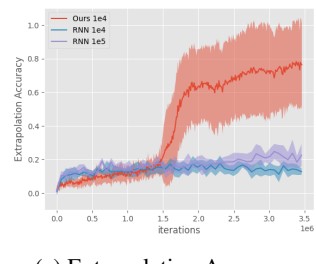

(a) Training Accuracy                (b) Test Accuracy                (c) Extrapolation Accuracy

*Figure 2.* **Generalization and Extrapolation** We train with a curriculum starting with 2-term expressions, expanding the dataset every 5e4 iterations until we have added 5-term expressions. Our method generalizes to mathematically and linguistically different 5-term expressions and extrapolates to 10-term expressions. Even with pre-training on auxiliary language translation tasks, the RNN does not extrapolate at all and only generalizes to equal-length but different language-pair expressions when given 10 times more data.

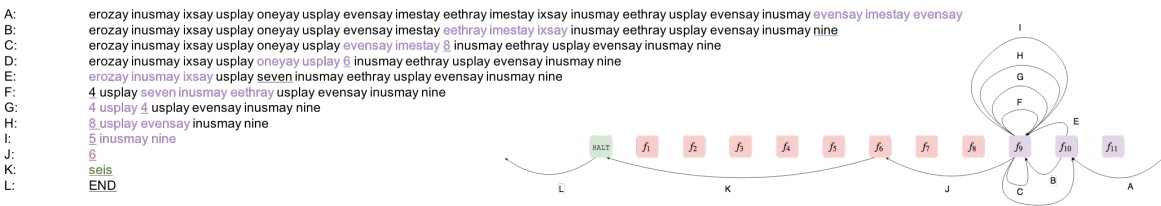

*Figure 3.* **Qualitative Analysis:** A visualization of a randomly selected execution trace from the extrapolation set. The input is the math expression $0 - 6 + 1 + 7 \times 6 - 3 + 7 - 7 \times 7$ expressed in Pig Latin. The desired output is *seis*, which is the value of the expression, 6, expressed in Spanish. Each step of computation is labeled A - L, which is reflected in the finite state machine view on the right. The purple functions are reducers and the pink functions are translators. The input to a function is highlighted and the output of the function is underlined. The controller learns order of operations by attending to where to apply the reducer. The reducer $f_9$ always reduces to numerals and reducer $f_{10}$ always reduces to English terms. This is interesting: because the functions do not know what the target language is, they learn to specialize for a specific purpose, resulting in hybrid arithmetic expressions made of different languages, the learner's own *internal* language. Note also that the learner exhibits *meta-reasoning*, or reasoning about its own computation: it has never seen the (Pig Latin, Spanish) language pair before during training, but it has seen (Pig Latin, Numerals) and (Numerals, Spanish), so it first reduces the Pig Latin expression to a numerical evaluation, and then translates that to its Spanish representation using the translator $f_6$. Note that all of this computation is happening internally to the learner, which computes on softmax distributions over the vocabulary; for visualization we show the token of the distribution with maximum probability.

sequence of softmax distributions over the vocabulary from another length $k$ sequence. We train on length-5 expressions. The test (length-5 expressions) and extrapolation (length-10 expressions) sets have different language pairs.

**Generalization and Extrapolation with Low Data:** Figure 2 shows that an RNN fails the translational and recursive challenges when given only $\sim 10^4$ training examples, but our method does not. This is challenging because the learner must reason with the knowledge of language pairs it has seen in order to solve unseen language pairs; it also needs to discover the order of operations as an invariant regularity, such that it can extrapolate to expressions of longer length by applying reducers in the right locations.

**Metareasoning:** Figure 3 shows a visualization of the execution trace of the learner on the extrapolation set which contains length-10 expressions in different language pairs, exhibiting both the translational and recursive challenges. The controller learns to reason about its own computations by routing through expressions it does know how to work

with to solve a new problem that it has never seen before. Our method learns specialized reducers that reduce to a particular language, which result in intermediate hybrid-language expressions. After it has reduced the entire expression, it takes an *additional* step to translate the final term to a term in the target language, even though the term before translation is not in either the source or the target language.

## 5. Discussion

We described a paradigm for learning to discover and exploit the structure of compositional problems. Our learner learns the structure and parameters of a program that dynamically customizes itself to the problem instance. Because its functions are also learned, our learner is not restricted to symbolic domains. Thus we believe our focus on compositionality is an important link for bridging the program induction community with other fields such as robotics. Future work will scale this approach to higher-dimensional input spaces.

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
