# OpenReview forum: "Automatically Constructing Compositional and Recursive Learners"
_ICML.cc/2018/Workshop/NAMPI — NAMPI 2018_

### Review · AnonReviewer3 · 2018-06-17
**Interesting direction but misses related work and sells a little bit too much**

**Rating:** 6
**Confidence:** 5

**Review:**

The paper considers the problem of automatically constructing compositional and recursive learners. That is, it consider the task of learning algorithmic procedures. The key parts are transformations of the representation and the sequential decision-making.

Overall, the paper is well written and touches a very interesting direction. However, there are some major downside, even for a workshop paper.


Although I very much like the direction of the paper, it is also a little bit to strong on the selling side. Consider e.g. the very introduction. The claim that a machine cannot learn how to stack blocks is not really true. Within the field of statistical relational learning, see e.g.

Luc De Raedt, Kristian Kersting, Sriraam Natarajan, David Poole (2016): Statistical Relational Artificial Intelligence: Logic, Probability, and Computation. Synthesis Lectures on Artificial Intelligence and Machine Learning, Morgan & Claypool Publishers, ISBN: 9781627058414,

there has been a number of works considering this task. Here recursion has played a major role (also in earlier work on inductive logic programming). In particular (for the present paper), work on relational reinforcement learning has considered exactly this task. Without discussing this line of research, the present paper is unfortunately giving the wrong impression that machine can not do that. Indeed, the work on relational reinforcement learning (RRL) does not mean that the task has been completely solved, but there you learn algorithmic procedures (in terms of logical programs) that even feature uncertainty, rewards and recursion. Also the idea of translational problem has been covered already in neuro-symbolic approaches (without so much on the deep part) where graph transformers have played a major role (see e.g. the work around Marco Gori).

Now, what is kind of new is the idea to use differentiable programs. This is indeed interesting as a growth path for the whole community. However, the experiment do not show that the proposed architecture solves the task set out in the introduction. The authors should also show results on longer arithmetic sequences. Only if we can really generalize to any length we would have solved the task. Alternatively the authors should lower the tone, i.e., make their statements (e.g. also in the conclusions) less general. This should also include a discussion of pros and cons. How much training examples will we need to solve the general tasks? Does this all really scale? How sensitive is the approach to parameter initialization? What are appropriate baselines? A neural network without recursion definitely will have a much harder time to solve the task.

To summarize, and interesting paper but it misses major related work, does not discuss downsides of the approach and also does not fully solve the task set out in the intro.
Given that NAMPI is a workshop, overall I am still leaning towards accept but only if the related work is appropriately discussed (not just some deep approaches but state of the art in general) and pros and cons are discussed.

---

> ### Comment · ~Michael_Chang1 · 2018-07-06
> **Response to AnonReviewer3**
>
> We thank Reviewer 3 for their careful and insightful feedback.
>
> Reviewer 3 is concerned about the claim in the introduction that a machine cannot learn to stack blocks and pointed out several related work. We thank Reviewer 3 for drawing our attention to this work and have included discussion on relational reinforcement learning. As far as we know, https://arxiv.org/abs/1709.10089 this is the state of the art for robotic learning for block stacking without any hardcoded priors; indeed it is true that robots can stack blocks with more prior information, but our intention was to highlight robotic block stacking without hardcoded priors.
>
> Reviewer 3 is concerned that our learner did solve the task set out in the introduction. This is a good point, and we have updated the introduction and conclusion to be more precise about our contribution. We would like to clarify that we intended the second sentence of the introduction merely for motivational purposes, to show the gap between the the current state of the art in robotic learning on block stacking without strong hardcoded priors (e.g. https://arxiv.org/abs/1709.10089), which, while impressive, still does not exhibit the same type of generalization that humans do. Our focus for that sentence is not to claim that robots cannot generalize in block-stacking if they have hardcoded primitive operations; rather our main focus is the problem of learning both the primitve operations and their rules of composition, which to the best of our knowledge a machine still cannot do in the context of block stacking.
>
> Reviewer 3 also brought up a good point about the pros and cons of our approach. We have updated the Future Work section accordingly to briefly discuss the advantages and disadvantages of our approach. In terms of sensitivity to parameter initialization, we would direct Reviewer 3 to our responses to Reviwer 1's questions (1) and (3).

---

### Review · AnonReviewer2 · 2018-06-21
**Good workshop submission**

**Rating:** 7
**Confidence:** 5

**Review:**

This is a clearly written paper about using a mix of RL and differentiable methods to learn both function application order and the functions themselves. The work is well motivated, and the model is clear. I concur with AnonReviewer3 that the paper sells itself a little hard given the fairly thin experimental results, but it's "just" a workshop and I think this is an excellent initial foray into this sort of technique. The paper should be presented at the workshop, but the authors should be mindful that more substantial experimental coverage would be needed to fully support the proposal, in follow up work.

---

> ### Comment · ~Michael_Chang1 · 2018-07-06
> **Response to AnonReviewer2**
>
> We thank Reviewer 2 for their careful and insightful feedback. We have updated the manuscript to be more precise on our contribution.

---

### Review · AnonReviewer1 · 2018-06-22
**Good for a workshop**

**Rating:** 7
**Confidence:** 4

**Review:**

The paper presents the idea to structure a neural network as (1) a set of learned reusable modules and (2) a controller that selects which module to call next. All components are neural, so the learned final algorithm is not expressed in interpretable source code. The hope is that modules will spontaneously specialize to individually perform modular operations that can be combined to solve difficult tasks. In the presented experiments these modules are interpretable at convergence (e.g. f6 is interpreted as a to-spanish translator).

Overall, I think the work is good and interesting enough to be accepted as a workshop submission.

Questions:
1. How does the training procedure break the symmetry between the different learnable modules. If I understand correctly, at the start of training, a set of "reducers" and a set of "translators" are randomly initialized. Is it the case that the random initialization is all that breaks symmetry to determine which function will eventually specialize to each language? Is training this system easy or does it sometimes get stuck in configurations where symmetry is not broken and the modules are not interpretable? Since a curriculum is used, I suspect that training is maybe not always easy.

2. I found the description of the architecture a little sparse. E.g. What is the architecture of the controller? Since the input varies in length, I guess it is an RNN? Some of the introductory remarks could be cut to fit in more details here.

3. What is the loss if "halt" is called too early and there are still multiple tokens in the final expression? Also, what happens if a reduction operator is called when there is only one token in the expression?

While the toy experiment is ok for a workshop submission, to make this work into a full conference submission, it will be key for the authors to show additional more realistic setting where their model can add value. As it currently stands, I don't think some of the claims in the introduction are not fully supported by the single experimental result.

Minor:
The caption of Figure 3 contains a typo: The Pig Latin expression reads
0 − 6 + 1 + 7*3*6 − 3 + 7 − 7*7

---

> ### Comment · ~Michael_Chang1 · 2018-07-06
> **Response to AnonReviewer1**
>
> We thank Reviewer 1 for their careful and insightful feedback.
>
> 1. Reviewer 1 asked about how the training procedure breaks the symmetry between the different learnable modules. We believe there are several factors that break the symmetry. Reviewer 1 is correct that the set of reducers and translators are randomly initialized. This is not the only sort of initial randomness; the controller is also randomly initialized as well. The training procedure for the controller follows the standard Proximal Policy Optimization training procedure, where the learner samples a set of episodes, pushes them to a replay buffer, and every k episodes updates the controller based on the episodes collected. Independently, every k' episodes we consolidate those k' episodes into a batch and use it to train the functions. Therefore, for episodes 0 through k the controller still has the same random initial weights, and for episodes 0 through k' the functions still have the same random initial weights. Because of the initial randomness, the initial controller will select certain functions more than others for certain inputs; similarly initially certain functions will perform better than others for certain inputs. Therefore, after k iterations, the controller's parameters will update in a direction that will make choosing the functions that luckily performed better for certain inputs more likely; similarly, after k' iterations, the functions' parameters will update in a direction that will make them better for the inputs they have been given. So gradually, functions that initially were slightly better at certain inputs will become more specialized towards those inputs and they will also get selected more for those inputs.
>
> Reviewer 1 also asked about the difficulty of training. In terms of training stability, we've observed that runs across all random seeds converge to solve the problems in the training set. Those solutions in general generalize to the testing or extrapolation set, but it is true that there is higher variance in performance, as shown in Figure 2. In Figure 2 we plot the median and the 10- and 90-percentiles (in our submission we had plotted mean and standard deviation, but for the camera ready it is median and percentile because that is more meaningful), showing that most runs consistently achieve close to 100% generalization, but not all do. Therefore, we believe that reducing the variance in generalization performance will require further work. We have indeed observed that a curriculum is tremendously helpful for the learner to converge in a timely manner; even just training on only five term multilingual expressions without the curriculum is at least five times as slow as training with the curriculum; the curriculum is also helpful in helping the functions "crystallize" into useful primitive transformations: without a curriculum we found that the learner was unable to generalize to the testing or extrapolation set.

---

> > ### Comment · ~Michael_Chang1 · 2018-07-06
> > **Response to AnonReviewer1 continued**
> >
> > 2. Reviewer 1 asked about the description of the architecture. The controller consists of a policy and a value function, each implemented as RNNs that read in the input expression. The value function outputs a value estimate for the current expression. Given the input expression, the policy first samples whether to halt, reduce, or translate, and then conditioned on that choice it samples the reducer or translator (if it doesn't halt). Given space constraints, we will elaborate on architectural details in a longer version of the paper which will upload on arxiv.
> >
> > 3. Reviwer 1 makes a good observation of two important nuances of the domain. In the case that "halt" is called to early, our learner treats it as a no-op. Similarly, if a reduction operator is called when there is only one token in the expression, the learner also treats it as a no-op. There are other ways around this domain-specific nuance, such as to always halt whenever halt is called, but only do backpropagation from the loss if the expression has been fully reduced (otherwise it wouldn't make sense to compute a loss on an expression that has not been fully reduced). The way we interpret these "invalid actions" is analogous to how people in RL sometimes handle the case when an agent in a maze walks into a wall, which is just to stay in the same state.
> >
> > Reviewer 1 mentioned that the claims in the introduction are not fully supported by the single experimental result. It is true that we did not solve the block-stacking task in the second sentence of the introduction, but this sentence was intended merely for motivational purposes, to show the gap between the the current state of the art in robotic learning on block stacking without strong hardcoded priors (e.g. https://arxiv.org/abs/1709.10089), which, while impressive, still does not exhibit the same type of generalization that humans do. Having set this type of generalization as our high-level target, we proceeded to distill the essence of the research problem. Multilingual arithmetic exhibits many of the properties of the type of compositional and recursive generaliztion we desire, and our result does demonstrate this type of generalization. We definitely agree that further work would be required to demonstrate the generality of our approach other domains, but we feel that the investigation conducted in this paper on the arithmetic domain does help us carefully study the problems we care about.
> >
> > Finally, Reviewer 1 is concerned that the experiment in this paper is too much of a toy problem. While we agree that we have not demonstrated our learner on a real world system, we would like to emphasize that our problem does display many of the challenges seen in real world problems, and has precedent; for example, papers in neural program induction still generally evaluate on simple algorithmic tasks such as sorting or addition. In our consideration, the multilingual arithmetic domain is already quite challenging because we learn both the structure and transformations of the program. However, we do share Reviewer 1's sentiment for scaling up our learner to more domains; this will be the subject for future work.

---

### Decision · ~NAMPI_Admin1 · 2018-06-28
**Paper2 Final Decision**

Accept

---

> ### Comment · ~Michael_Chang1 · 2018-07-06
> **Revision after reviews**
>
> We thank all reviewers again for their thoughtful and in-depth reviews. The reviewers helped us improve the paper in several dimensions, and the revised manuscript is uploaded.